# Validation of Cognitive Rehabilitation as a Balance Rehabilitation Strategy in Patients with Parkinson’s Disease: Study Protocol for a Randomized Controlled Trial

**DOI:** 10.3390/medicina57040314

**Published:** 2021-03-26

**Authors:** Aida Arroyo-Ferrer, Francisco José Sánchez-Cuesta, Yeray González-Zamorano, María Dolores del Castillo, Carolina Sastre-Barrios, Marcos Ríos-Lago, Juan Pablo Romero

**Affiliations:** 1Facultad de Ciencias Experimentales, Universidad Francisco de Vitoria, Pozuelo de Alarcón, 28223 Madrid, Spain; aida.arroyo@ufv.es (A.A.-F.); fjose.sanchez@ufv.es (F.J.S.-C.); y.gonzalezz@alumnos.urjc.es (Y.G.-Z.); 2Escuela Internacional de Doctorado, Department of Physical Therapy, Occupational Therapy, Rehabilitation and Physical Medicine, Universidad Rey Juan Carlos, Alcorcón, 28933 Madrid, Spain; 3Neural and Cognitive Engineering Group, Centre for Automation and Robotics, Spanish National Research Council—Arganda del Rey, 28500 Madrid, Spain; md.delcastillo@csic.es; 4Neuronup SL, 26006 Logroño, Spain; carolina@neuronup.com; 5Basic Psychology II Department, UNED, 28040 Madrid, Spain; mrios@psi.uned.es; 6Brain Damage Unit, Hospital Beata María Ana, 28007 Madrid, Spain

**Keywords:** Parkinson’s disease, balance, information processing speed, attention, cognitive rehabilitation

## Abstract

*Background*: Parkinson’s disease (PD) is the second most common neurodegenerative disorder. This disease is characterized by motor symptoms, such as bradykinesia, tremor, and rigidity. Although balance impairment is characteristic of advanced stages, it can be present with less intensity since the beginning of the disease. Approximately 60% of PD patients fall once a year and 40% recurrently. On the other hand, cognitive symptoms affect up to 20% of patients with PD in early stages and can even precede the onset of motor symptoms. There are cognitive requirements for balance and can be challenged when attention is diverted or reduced, linking a worse balance and a higher probability of falls with a slower cognitive processing speed and attentional problems. Cognitive rehabilitation of attention and processing speed can lead to an improvement in postural stability in patients with Parkinson’s. *Methods*: We present a parallel and controlled randomized clinical trial (RCT) to assess the impact on balance of a protocol based on cognitive rehabilitation focused on sustained attention through the NeuronUP platform (Neuronup SI, La Rioja, Spain) in patients with PD. For 4 weeks, patients in the experimental group will receive cognitive therapy three days a week while the control group will not receive any therapy. The protocol has been registered at trials.gov NCT04730466. *Conclusions*: Cognitive therapy efficacy on balance improvement may open the possibility of new rehabilitation strategies for prevention of falls in PD, reducing morbidity, and saving costs to the health care system.

## 1. Introduction

Parkinson’s disease (PD) is the second most common neurodegenerative disorder. This disease is characterized by motor symptoms, such as bradykinesia, tremor, and rigidity. Non-motor symptoms such as cognitive impairment, anosmia, sleep disorders, or depression are also part of the disease, and although their prevalence is very high, they are often underdiagnosed [1,2].

One of the cognitive characteristics in PD is the slowness in the processing of information, which includes deficits in processing speed and attention, cognitive inflexibility, and forgetfulness [3]. These symptoms may appear from the initial stages of the disease [4].

Approximately 60% of PD patients fall once a year and 40% do so regularly [5]. These falls may be correlated with the inability to achieve compensatory movements to regain balance when their center of gravity generally oscillates outside their limits of stability (LOS), which is reduced in this disease [6]. Some authors point out that reaction times and processing speed [7] may be a marker of postural instability since a reduced speed is associated with difficulty in making turns [8]. This is in line with Pantall’s findings, indicating that cognitive function and postural control progressively worsen with disease progression [9].

The relationship between cognitive impairment and postural instability in PD patients may be specific for tasks that assess the dorsolateral prefrontal cortex and its frontal–subcortical connections [10]. The main cognitive functions whose affectation would influence a worse balance and gait performance would be attention and executive functions [7,11,12,13]. Varalta et al. [12] specified that balance is related to executive functions and attention, while functional mobility is related to cognitive impairment, verbal fluency, and attentional capacity.

Some authors point out that within the executive functions the component with the greatest weight in this relationship would be the inhibitory control [13]. Dual-task performance has also been established as a good indicator of falls in patients with early-stage PD and no previous history of falls [14]. The studies that carried out a one and a half years follow-up of the participants concluded that the deterioration of executive functions acts as a predictor of future falls in patients with PD [7,9,15,16].

Cognitive rehabilitation through neurorehabilitation platforms and neuropsychological rehabilitation in patients with Parkinson’s disease has shown to be effective in improving processing speed, attention, and executive functions [17,18,19,20].

Although the relationship between cognitive deficits and postural stability seems to be demonstrated [21] we have not found studies that, through cognitive rehabilitation, seek a stability improvement.

The main aim of this study is to conduct a parallel and controlled randomized clinical trial to validate a rehabilitation protocol for sustained attention and information processing speed in patients with Parkinson’s disease, targeting changes in 1. instrumental and clinical evaluation of postural stability and 2. wide neuropsychological and neurophysiological evaluation. Our main hypothesis is that the group that receives cognitive rehabilitation will improve their postural stability compared to the group that does not undergo any therapy. If this hypothesis is confirmed it will open the possibility of new rehabilitation strategies for prevention of falls in PD, reducing morbidity, and saving costs to the health care system.

## 2. Materials and Methods

Mat the Standard Protocol Items: Recommended Items for Interventional Trials (SPIRIT) 2013 checklist has been used to assure the quality of the protocol [22].

### 2.1. Study Design and Participants

The participants will be recruited in PD outpatient’s clinic of the Beata María Ana Hospital of Madrid and referred from other collaborators centers and hospitals or self-referrals that know the research project due to the dissemination through social networks. The subjects included will be assessed by a neurologist expert in movement disorders (J.P.R.)) (Figure 1). All patients will be idiopathic PD patients diagnosed according to the UK Parkinson’s Disease Society Brain Bank criteria. The inclusion and exclusion criteria are shown in Table 1. During the protocol, changes in dopaminergic medication will be prohibited and, if necessary, will lead to the exclusion of the patient. The total daily dose of dopaminergic drugs will be recorded and controlled in the final analysis of the results.

To control the Wearing-off effect, a specific questionnaire will be applied both at the beginning and at the end of the study [23].

The data will be anonymized and will be recorded separately, being safeguarded in accordance with current European data protection laws. All data will be recorded and verified twice in a database designed for the study.

The study design will be a parallel, randomized and controlled experimental study. The patients included in the sample will be divided into two groups: Cognitive rehabilitation (experimental) and no therapy (control). The distribution will be random, but it will be ensured that there is an equitable number of subjects in each of the stages of the disease, being matched by their score in HY. The randomization of the sample will be done through the website: http://www.randomization.com/ (accessed on 12 February 2020). The duration of the study will be 18 months.

The sample will be composed of 46 patients randomly distributed in a 1:1 ratio (Figure 1). To calculate the sample size, we used the GRANMO calculator. Accepting an alpha risk of 0.05 and a beta risk of 0.2 in a one-sided contrast, 23 subjects are required to detect a difference equal to 12 points in the Biodex objective stability limits test assessment (Biodex, Version 1.08, Biodex, Inc., Shirley, NY, USA). The standard deviation in the stability limits test obtained in our sample from the NeuroMOD project was 16.16. Considering a 15% loss, it will be necessary to reach a sample of 26 patients in each group.

### 2.2. Intervention Protocol

The intervention sessions will be held at the patient’s home. If this is not possible, they will be carried out in the research laboratory of the Beata María Ana Hospital or in the Parkinson’s patients association or current attending hospital of the patient.

Depending on the assignment to each of the experimental groups, the patients will undergo neurorehabilitation of cognitive functions or will not receive any therapy.

Due to the characteristics of the study, all participants will know the experimental group to which they belong. The researcher evaluating the preintervention and postintervention balance is blinded to the experimental group.

Control Group: The control group will not receive any therapy. They will be simply evaluated at the same time as the experimental group.

Experimental Group: In the experimental group, patients will receive a 12-session neuropsychological rehabilitation protocol that will be carried out over four weeks (3 weekly sessions). The protocol and the number of sessions has been designed by neuropsychologists following the Díez-Cirarda et al. recommendations [24] (Figure 1 and Figure 2).

Each session will consist of a series of cognitive stimulation tasks aimed at improving attention, perception, and processing speed. These tasks will be applied through the NeuronUp (Neuronup SI, La Rioja, Spain) cognitive neurorehabilitation platform to achieve maximum homogeneity and objectivity in their application. The protocol is preprogrammed in the application and is identical for all patients. Performing tasks through the platform allows one to objectively record the performance of each patient in all tasks to ensure adherence and efficacy. Before the first session, a training session will be carried out under the supervision of a researcher. All doubts that may arise will be resolved.

The NeuronUp (Neuronup SI, La Rioja, Spain) cognitive neurorehabilitation platform has a home activities module. In this case, the tasks and the days of their completion will be assigned, and the patient has to access the platform in a computer connected to internet to perform the intervention. The program has a direct researcher–patient feedback tool, in which the intervention and evolution of the patient can be followed in real time, being able to adjust the contents in the same way as in person. The researcher will be in direct contact with the patient and will remind the patient of the day and time at which he should carry out the session and will check at the end of the session that it has been carried out successfully.

During the intervention, the subjects will be seated in a relaxed position in front of the computer. No adverse effects are expected, in case of occurrence, mild adverse effects due to the use of the computer such as dizziness, mental fatigue, headaches, or other will be evaluated after each session if they persist, they will be notified to a licensed physician who can decide the management of the patient.

### 2.3. Outcomes Measurement

All detailed measures will be evaluated in the two groups before and after the intervention. To ensure that all patients are in the “ON” state, the evaluation and intervention will take place one hour after taking the dopaminergic medication and UPDRS III [25] will be recorded before the measurements.

The evaluation prior to the intervention will be carried out in a single session on the Thursday or Friday before the start of the intervention and the subsequent one, on Monday or Tuesday after the end of the intervention. Necessary breaks can be taken to avoid patient fatigue (Figure 2).

#### 2.3.1. Main Outcomes

##### Instrumental Stability Evaluation

Objective metrics using the Biodex Biosway portable balance system [26] (Biodex, Version 1.08, Biodex, Inc., Shirley, NY, USA) that allows one to determine the following determinations:

1. Stability limits test: This test is a good indicator of dynamic stability control with a normalized balance of improvement [27]. The patient shifts its weight over a platform to control the cursor displacement in order to reach targets in a screen. The position of the objectives has been pre-established by the manufacturer at 50% of the stability limits according to the height of each volunteer. The test dependent variables provided by the machine are the time (seconds) it takes for the subject to complete the test and the directional control value, which is the proportion of the distance travelled by the cursor from the center to each target (based on that 100% is a straight line from the center to the intended target). This means that the greater the distance travelled by the subject, the worse their stability.

2. Fall risk test: The test protocol is incorporated in this system and normative data is used to assess the risk of falls for the subjects.

##### Clinical Stability Evaluation


*Berg Balance scale*


The Berg balance scale (BBS) is designed to measure changes in functional standing balance over time. This scale measures the balance skills observed during tasks that involve sitting, standing, and changing positions. Total scores are indicative of general balance skills [28].


*Timed stand up and go test*


The timed “Up and Go” test [29] is a common mobility measure in rehabilitation. This test measures parameters such as turning, sitting, and walking speed. The mean angular velocity during turning and the duration of the turning and sitting phase have been shown as valid measures of balance in PD and are also responsive to rehabilitation [30,31].

##### Neurocognitive Evaluation

A protocol designed to measure processing speed, sustained and alternating attention, working memory, interference control, visual search, and verbal fluency (Stroop test [32,33,34], Trail Making Test [35,36], WAIS-III Processing Speed-Digit Symbol Coding, Processing Speed-Symbol Search, Digit Span [37], FAS Word Fluency [38], and The Bells Test [39]) will be administered by a neuropsychologist (A.A.-F.) experienced in their application in order to evaluate the efficacy of the rehabilitation protocol.

##### Computerized Reaction Times Battery

Reaction time tests are a predesigned set of computerized tasks to objectively measure various cognitive domains such as perception and alertness, inhibition, and visual search. The system has been described in and published in Arroyo et al. [40].

#### 2.3.2. Secondary Outcomes

*Parkinson’s Disease Questionnaire 39* (PDQ39) [41]: It is a specific questionnaire for evaluating the quality of life of Parkinson’s patients. The quality of life of patients with PD is associated with the subject’s perception of their gait and balance. The improvement in mobility and balance is related to the improvement in the PDQ-39 scores, especially in the set of items that measure mobility (items 1–10) [42].

##### Neurophysiological Evaluation

The balance is related to an increase in theta power in the frontal, central, and parietal regions. Increased theta power in these zones correlates with better balance performance [43].

A resting state 64 electrodes EEG recording will be performed using the actiCHamp amplifier (Brain Vision LLC., Morrisville, NC, USA) with Ag/AgCl active scalp electrodes (actiCAP electrodes, Brain Vision LLC., Morrisville, NC, USA), acquisition will be carried out using NeuroRT Studio software (Mensia Technologies SA, Paris, France). The EEG signal processing procedure will be performed using MATLAB functions (MathWorks Inc., Natick, MA, USA), specifically the EEGLab toolbox. The spectral entropy correlation, the coherence, and the differences of interhemispheric divergence between the experimental group and the controls will be analyzed as results.

### 2.4. Data Analyses

The SPSS software package (version 25.00; SPSS Inc., Chicago, IL, USA; IBM, New York, NY, USA) will be used by a blinded statistician for statistical analysis of the data. Demographic and clinical variables will be presented as mean and standard deviation. A 95% confidence interval (CI) will be taken, considering statistically significant all those values that had a *p* < 0.05. Parametric statistical tests will be used for hypothesis testing (*n* > 30). To show the differences in the variables, a 2 × 2 analysis of variance (ANOVA) of repeated measures will be carried out, taking time and group as factors, being the control and experimental groups and the pretreatment and posttreatment time measures. If significant differences are found in any of the interactions, a post hoc analysis with Bonferroni correction will be performed. The effect sizes will be calculated using Cohen’s d, classifying them into small (d between 0.20 and 0.49), medium (d between 0.50 and 0.79), or large (d greater than 0.80) following Cohen’s method.

Possible confounders or mediating variables will be included as covariates in the analysis.

For the rest of the variables, a Pearson correlation or linear regression will be carried out with the motor and balance variables. The low correlation will be that with values between 0.2 and 0.39; the correlation will be considered moderate when the value goes from 0.40 to 0.79; and high when the correlation value is greater than 0.80.

For the EEG entropy analysis, nonparametric Kruskal–Wallis (KW) tests will be performed to compare the distributions, the Mann–Whitney U test to evaluate the difference between the median values, and the Wilcoxon rank sum test for measure pre/post intervention changes, with a significance level of 0.05 and the alternative hypothesis of unequal values. Therefore, small *p*-values will suggest significant differences between populations.

### 2.5. Dissemination Plans

All results will be published in specialized scientific journals. The final data of these studies will be anonymized and will remain accessible after their publication and may justifiably be requested from the main inventors. The results will be made public through the social networks of our institution.

## 3. Discussion

The main objective of the study is to improve the balance of Parkinson’s patients through cognitive rehabilitation. Cognitive rehabilitation in PD has been effective in the rehabilitation of attention and executive functions [44], on the other hand, previous studies relate attention and executive functions with the risk of falls, balance, and gait in PD patients, proposing rehabilitation in these areas as a possible effective therapy [8,45].

All the clinical variables assessed will allow one to identify markers that predict a better outcome of the cognitive intervention on balance rehabilitation, this information will be of great relevance to adjust the intervention to specific profiles of patients with PD who will benefit the most.

Although our study counts with the limitation of a lack of blinding of the participants and some researchers, its design is robust enough to provide with evidence that may support new approaches to stability rehabilitation in PD. Current strategies continue to be based on training in compensatory and signaling strategies, high intensity visual rehabilitation, and exercise such as dance, Tai-Chi, and Yoga [46]. Other studies use virtual reality as part of therapy although there is still no evidence on its efficacy [47]. The use of virtual reality implies the transfer of patients to a specialized center and continuous supervision. Our study proposes a home-based protocol that can improve the quality of life of Parkinson’s patients even from their own home avoiding the current pandemic limitations on medical care.

## 4. Conclusions

Cognitive therapy through neurorehabilitation platforms opens up the possibility of new rehabilitation strategies for the prevention of falls in PD, reducing morbidity, and saving costs for the health system.

## Figures and Tables

**Figure 1 medicina-57-00314-f001:**
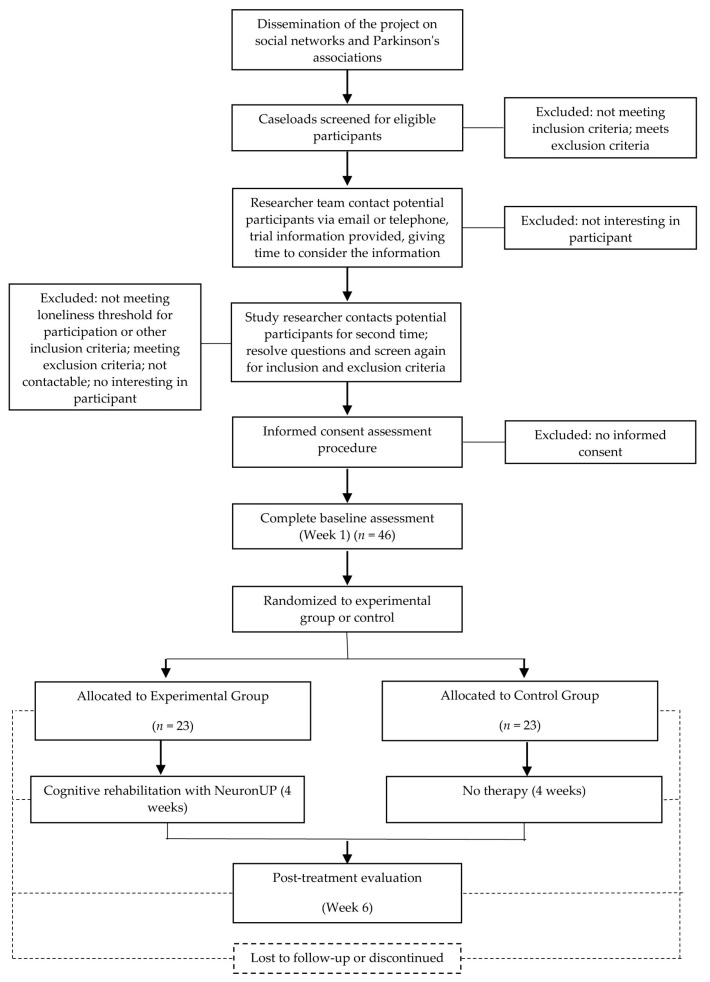
Standard Protocol Items: Recommended Items for Interventional Trials (SPIRIT) study flow chart.

**Figure 2 medicina-57-00314-f002:**
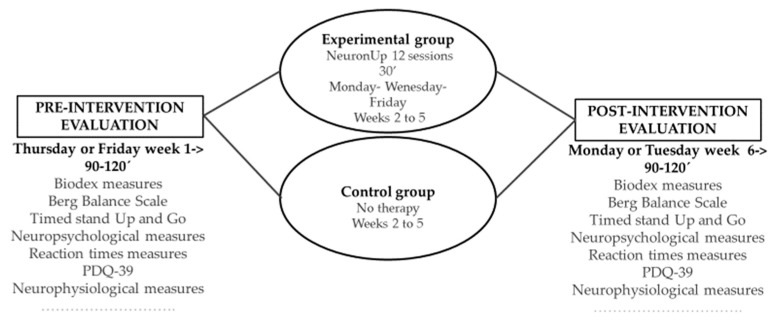
Schematic representation of the study experimental procedures.

**Table 1 medicina-57-00314-t001:** Inclusion and exclusion criteria.

Inclusion Criteria	Exclusion Criteria
Idiopathic Parkinson’s disease (diagnosed according to the UK Parkinson’s Disease Society Brain Bank criteria). Older than 18. Stage < III Hoehn–Yahr with no obvious motor fluctuations.	Visual-perceptual difficulties. Peripheral sensory disturbances due to polyneuropathy. Cerebellar alterations. Impossibility to operate a computer due to motor impairment Severe cognitive impairment (MoCA < 24). Moderate or severe active depression (BDI > 14). Dependence (mRS > 3). Dopaminergic medication changes in the last 30 days. Structural changes MRI Severe comorbidity (cancer, severe COPD, etc.). Atypical data for idiopathic PD.

## Data Availability

The data presented in this study will be available on request from the corresponding author.

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
