# Peer review of "Validation of Cognitive Rehabilitation as a Balance Rehabilitation Strategy in Patients with Parkinson’s Disease: Study Protocol for a Randomized Controlled Trial"

_medicina, 2021, doi:10.3390/medicina57040314_

Round 1

Reviewer 1 Report

In material and methods. 

2.2 protocol intervention: the authors should explain more precisely whether the patients conducted the training protocol alone at home or under a supervision of a qualified person. 

If they were not followed in each training session, how could they verify the riliability of the administered protocol?

Author Response

Thank you very much for your time reviewing our manuscript.

Training in the program will take place under supervision in the first session. The sessions will be carried out by the patients from home, but they will have direct feedback from the researcher. The program is the same for everyone, as is the time of completion of each task, once it is finished, a log is generated and can be checked by the researcher in the same program.

Clarifications have been included in the manuscript, lines 190-191 and 197-200

Reviewer 2 Report

I believe the results of the are consistent with the study design. the design is original however future research should expand the sample size. 

Author Response

Thank you very much for your comment. Regarding the sample size, we have used the GRANMO effect size calculator to calculate the number of subjects necessary to find differences between the two groups in postural stability. Nevertheless, we will not close the recruitment when we reach 46 subjects, we will expand the sample while the necessary resources for the study are available.

Reviewer 3 Report

It is difficult to review a manuscript for a study that hasn't been conducted.  There are no data, analyses, discussion points to comment on. 

From a methodological perspective, we will share a few considerations:

1. The authors might consider matching groups by motor phenotype (tremor dominant vs non-tremor dominant) as the latter group is more likely to experience falls and is at greater risk for developing more severe cognitive impairment. 

2. Lack of blinding should be reconsidered.

3. How will the authors account for the the impact of motor impairment on participant computer access and reaction times?

4. Instead of a no-therapy group, the authors may wish to consider a standard rehabilitation group. 

Author Response

Thank you very much for reviewing our manuscript.

  1. The authors might consider matching groups by motor phenotype (tremor dominant vs non-tremor dominant) as the latter group is more likely to experience falls and is at greater risk for developing more severe cognitive impairment. 

Thank you for your comment. The phenotypes differences pointed are relevant, but in our knowledge, although single domain non-amnesic mild cognitive impairment has been related to postural instability and gait disorder, it currently believed to be mediated by the Hoehn & Yahr scale score and current cognitive status (1–5).

Our study's inclusion criterion is a score on the HY scale <III and MoCA >24, for which we do not expect differences between both phenotypes.  UPDRS allows us to register the phenotype of each patient and we can segment the sample after completion of the study to compare the performance not only in basal balance assessment but also in their response to the intervention.

  1. Lack of blinding should be reconsidered.

Thank you for the suggestion, we have modified the blinding in the study (manuscript, lines 169-171).  The researcher who evaluates postural stability, with the BIODEX platform, timed stand up and go test and Berg balance scale, will be blinded to the experimental group.

  1. How will the authors account for the impact of motor impairment on participant computer access and reaction times?

Reaction times have been used for multiple pathologies (6,7), being effective in being able to evaluate cognitive processes objectively. In our previous studies on Parkinson's disease, we have not found motor impairments to limit the operation of a computer in any patient (8).

Not being able to perform reaction times pressing a key would result in the inability to perform cognitive training and the patient would be excluded. This reason for exclusion has been included in Table 1: inclusion and exclusion criteria

Motor impairment is controlled by the finger tap task that considers just the motor execution of the key tapping and can be used to control the results of more complex tasks.

  1. Instead of a no-therapy group, the authors may wish to consider a standard rehabilitation group. 

Our objective is not to compare two therapies (physiotherapy stability rehabilitation vs cognitive rehabilitation), but to see if cognitive therapy is effective in improving postural stability and balance. We cannot assure that physiotherapy has some cognitive unintentional effects on improving attention. In the future, the comparison of the two therapies could be considered, but always including a no-therapy group.

References

  1. Sollinger AB, Goldstein FC, Lah JJ, Levey AI, Factor SA. Mild cognitive impairment in Parkinson’s disease: Subtypes and motor characteristics. Parkinsonism Relat Disord. 2010 Mar;16(3):177–80.
  2. Herman T, Weiss A, Brozgol M, Wilf-Yarkoni A, Giladi N, Hausdorff JM. Cognitive function and other non-motor features in non-demented Parkinson’s disease motor subtypes. J Neural Transm. 2015 Aug 1;122(8):1115–24.
  3. Marras C, Chaudhuri KR. Nonmotor features of Parkinson’s disease subtypes. Mov Disord Off J Mov Disord Soc. 2016 Aug;31(8):1095–102.
  4. Goldman JG, Weis H, Stebbins G, Bernard B, Goetz CG. Clinical differences among mild cognitive impairment subtypes in Parkinson’s disease. Mov Disord. 2012;27(9):1129–36.
  5. Cholerton BA, Zabetian CP, Wan JY, Montine TJ, Quinn JF, Mata IF, et al. Evaluation of mild cognitive impairment subtypes in Parkinson’s disease. Mov Disord. 2014;29(6):756–64.
  6. Lubrini G, Ríos Lago M, Periañez JA, Tallón Barranco A, De Dios C, Fernández-Fournier M, et al. The contribution of depressive symptoms to slowness of information processing in relapsing remitting multiple sclerosis. Mult Scler J. 2016 Oct;22(12):1607–15.
  7. DeLuca J, Kalmar JH. Information Processing Speed in Clinical Populations. Psychology Press; 2013. 318 p.
  8. Components determining the slowness of information processing in parkinson’s disease - Arroyo - - Brain and Behavior - Wiley Online Library [Internet]. [cited 2021 Mar 10]. Available from: https://onlinelibrary.wiley.com/doi/full/10.1002/brb3.2031

Round 2

Reviewer 3 Report

I would be happy to review this manuscript once the study is conducted. I have indicated this to the Editor. 

Author Response

Dear Reviewer it will be a pleasure having you as a reviewer of the final results of our study. 

Thank you for your time and contributions to our protocol.